# Complete structure of the chemosensory array core signalling unit in an *E. coli* minicell strain

Alister Burt[1,6], C. Keith Cassidy [2,6], Peter Ames [3], Maria Bacia-Verloop[1], Megghane Baulard[1], Karine Huard[1], Zaida Luthey-Schulten [4], Ambroise Desfosses[1], Phillip J. Stansfeld [2], William Margolin [5], John S. Parkinson [3] & Irina Gutsche [1*]

Motile bacteria sense chemical gradients with transmembrane receptors organised in supramolecular signalling arrays. Understanding stimulus detection and transmission at the molecular level requires precise structural characterisation of the array building block known as a core signalling unit. Here we introduce an *Escherichia coli* strain that forms small minicells possessing extended and highly ordered chemosensory arrays. We use cryo-electron tomography and subtomogram averaging to provide a three-dimensional map of a complete core signalling unit, with visible densities corresponding to the HAMP and periplasmic domains. This map, combined with previously determined high resolution structures and molecular dynamics simulations, yields a molecular model of the transmembrane core signalling unit and enables spatial localisation of its individual domains. Our work thus offers a solid structural basis for the interpretation of a wide range of existing data and the design of further experiments to elucidate signalling mechanisms within the core signalling unit and larger array.

[1] Institut de Biologie Structurale, Université Grenoble Alpes, CEA, CNRS, IBS, 71 Avenue des martyrs, F-38044 Grenoble, France. [2] Department of Biochemistry, University of Oxford, South Parks Road, Oxford OX1 3QU, UK. [3] School of Biological Sciences, University of Utah, Salt Lake City, UT 84112, USA. [4] Department of Chemistry, University of Illinois Urbana-Champaign, Urbana, IL 61801, USA. [5] Department of Microbiology & Molecular Genetics, The University of Texas Health Science Center at Houston, Houston, TX 77030, USA. [6] These authors contributed equally: Alister Burt, C. Keith Cassidy. *email: irina.gutsche@ibs.fr

Bacteria survive and proliferate by sensing changes in their environment and responding through metabolic adaptation or locomotion[1,2]. Chemotactic bacteria, for example, monitor attractant and repellent concentration gradients and promote movement towards favourable niches. The chemosensory arrays that mediate this behaviour are assembled from core-signalling units (CSUs) that contain two trimers of receptor dimers (ToDs) interconnected at their cytoplasmic tips by a dimeric histidine autokinase CheA and two copies of a coupling protein CheW, which links CheA activity to receptor control (Fig. 1). Chemoeffector binding to the periplasmic domain of the receptors triggers a signalling cascade of intracellular phosphorylation events, which ultimately regulate the direction of the cell's flagellar motors. To allow appropriate sensory responses over a wide range of chemoeffector concentrations, receptor sensitivity is continuously tuned through the reversible methylation of receptors, known as methyl-accepting chemotaxis proteins or MCPs. *Escherichia coli* has four canonical MCPs that share a common functional architecture (Fig. 1a); the two numerically predominant ones are Tar (aspartate and maltose sensor) and Tsr (serine and autoinducer 2 sensor). Changes in the occupancy of MCP periplasmic ligand-binding domains trigger conformational rearrangements that propagate through the inner membrane to the HAMP domain (a signalling element which couples extracellular input to intracellular output in most microbial chemoreceptors and sensory kinases)[2] and then through an extended four-helix cytoplasmic domain[1]. The latter comprises three signaling regions: (1) a methylation-helix (MH) bundle, which contains the sites of reversible modification, (2) a flexible region containing the glycine hinge[3], and (3) the kinase-control region that binds CheA and CheW and enables receptor

trimerisation (Fig. 1a). Sensory signals are then transmitted from the receptor tips through CheA and CheW to affect CheA activity. Each of CheA's five domains, referred to as CheA.P1–P5 (Fig. 1b), plays a specific role in the autophosphorylation reaction: CheA.P5 facilitates regulatory coupling through receptors and CheW, CheA. P4 hydrolyses ATP to produce transferable phosphoryl groups, CheA.P3 enables dimerisation, and CheA.P1 and CheA.P2 mediate the transfer of phosphoryl groups from CheA.P4 to downstream response regulators.

Decades of work have given rise to detailed mechanistic models of signal transduction in MCPs[1,2], especially within individual receptor homodimers, and more recently, multiple CheA kinase-signaling models have been put forth[4–8]. While these models provide numerous insights and testable predictions, they are mainly limited to specific signaling modules or domains and/or lack residue-level detail. Moreover, owing to the complexity of the CSU assembly, the functional coupling between proteins, both within and between CSUs, remains poorly understood at the molecular level[8]. Progress towards a comprehensive signal transduction model has been hindered, in part, by the lack of an experimentally determined structure for a complete CSU, even at low resolution. On one hand, cryo-electron tomography (cryo-ET) structures of complexes composed of receptor cytoplasmic domains, CheA and CheW that are assembled on lipid mono-layers provide insights into CSU structure[9], but lack the ligand-binding, transmembrane, and HAMP portions of the receptor. On the other hand, cryo-ET maps of intact receptor arrays in situ thus far lack this level of detail and become diffuse at membrane-proximal regions, purportedly due to intrinsic flexibility of the signalling domains[10–13].

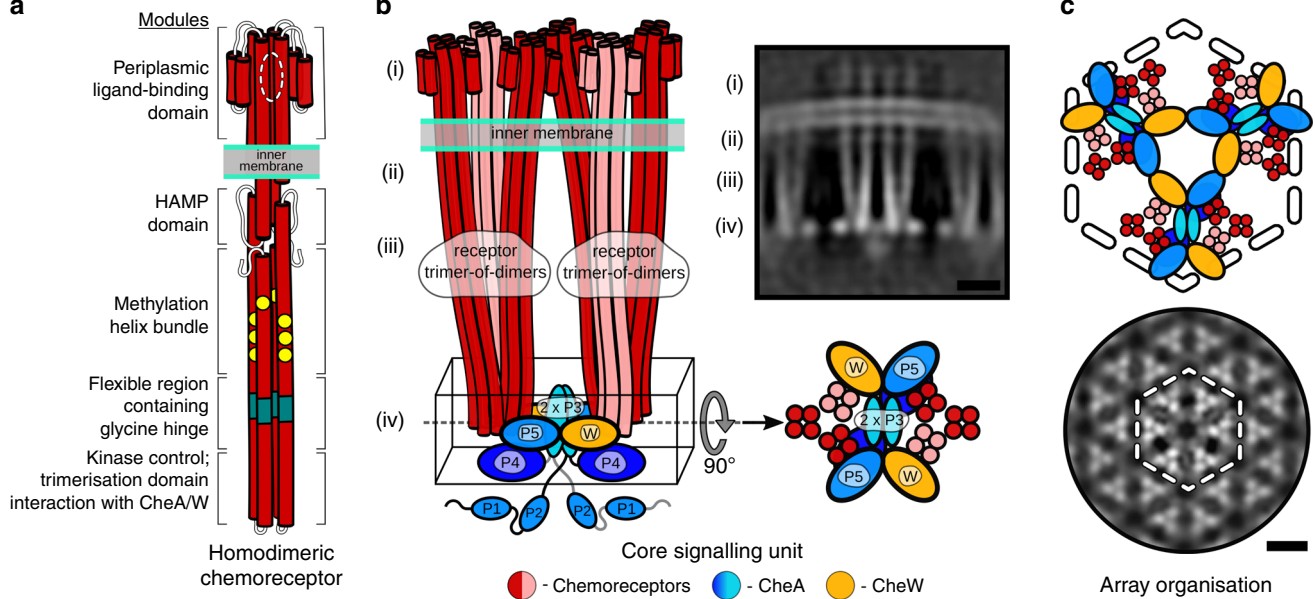

**Fig. 1 Overview of chemoreceptor, CSU and chemoreceptor array architectures. a** Schematic representation of homodimeric chemoreceptor structure. Red cylinders represent α-helical secondary structures drawn approximately to scale, flexible hinges are drawn as thin wavy strings, important regions discussed further in the text are highlighted (methylation sites as yellow circles, glycine hinges as teal cylinders). Regions encompassed in square brackets are the periplasmic ligand-binding domain (PP), the HAMP domain (HAMP), the methylation-helix bundles (MH), a flexible region containing the glycine hinge (GH) and the trimerisation and kinase control domain (KC) which is the site of interaction with CheA and CheW. **b**—left Two receptor ToDs interact with CheA and CheW to form a CSU shown from the side. Two MCP dimers in the CSU are shown in salmon for perspective. CheA is shown in shades of blue, and CheW in gold. CheA.P3, CheA.P4 and CheA.P5 are labelled and have known positions. Positions of CheA.P1 and CheA.P2 are not certain. The baseplate region is boxed. **b**—top right Same as in **b**—left shown as a 5 nm-thick projection through the density of our reconstruction. Roman numerals in **b**—left and **b**—top right refer to important regions of the structure—(i) the periplasmic domain, (ii) the HAMP domain, (iii) the methylation-helix bundle and (iv) the baseplate region. **c**—top CheA and CheW from three neighbouring CSUs interact to form chemoreceptor arrays, shown schematically and **c**—bottom as a 5 nm projection through the density of our reconstruction. The same region in **c**—top and **c**—bottom is delimited by a dashed hexagon. In **b**—top right and **c**—bottom, protein density is shown in white and scale bars are 10 nm. Source data are provided as a Source Data file.

While cryo-ET offers the best prospects for understanding the structures of receptors and their signalling arrays[14–16], most bacteria are not ideal candidates for direct cryo-ET imaging because their thickness exceeds the practical limit of ~500 nm. In principle this limit can be circumvented by vitreous sectioning[17,18] or focused ion beam milling[19–21]; however, these methods are expensive, technically demanding, and introduce the additional challenge of identifying sections containing the areas of interest. Bacterial minicells offer a valuable solution to the thickness problem. Minicell-producing strains frequently divide close to one pole of a rod-shaped bacterium to generate small achromosomal cells capable of normal metabolic functions. Such minicells are thus particularly useful for studies of polar regions of the bacterium, precisely where *E. coli* chemoreceptors organize into arrays[22].

Here we present the WM4196 minicell-producing *E. coli* strain from which we isolated small, healthy-looking minicells suitable for high-resolution analysis of internal structures by cryo-ET and subtomogram averaging. We observe that these cells contain extended, ordered and functional chemosensory arrays and therefore can be used to revisit the structure of the CSU. The resulting three-dimensional (3D) map enables us to propose a molecular model of the transmembrane CSU, which opens up avenues for further research.

## Results

**Initial characterisation of the WM4196 minicells.** Typical minicell preparations reveal numerous defects including stripped-off membranes, vesicles, spheroplasts, concentric membrane layers, and a markedly swollen periplasm[10,11,23]. The WM4196 strain produced a higher proportion of uniform and round minicells, many of which were <0.4 μm in diameter (Fig. 2a). In addition, cryo-ET reconstructions revealed flattening of these cells in vitreous ice, presumably caused by the plunge freezing process. This flattening reflects the relative plasticity of the cells and makes the sample thinner and, therefore, more suitable for high-resolution

cryo-ET analysis (Fig. 2b). Moreover, we discovered that about 50% of tomographic reconstructions of WM4196 minicells featured extensive chemosensory arrays (Fig. 2c), in contrast to a previous minicell study in which <20% contained visible arrays[11]. We therefore decided to characterise the WM4196 strain genetically and biochemically as a potentially useful tool for the structural investigation of chemosensory arrays.

Strain WM4196 is a distant derivative of a minicell mutant first described over 50 years ago[24]. Sequence analysis of the *minCDE* region in WM4196 revealed a G262D substitution within the MinD protein, which is presumably sufficient to abolish MinD function. MinD is normally required for the pole-to-pole oscillation of its partner protein MinC, which in turn prevents the essential FtsZ protein from forming cytokinetic rings at cell poles, limiting them to proper mid-cell location where they divide cells by binary fission[25]. In cells without a functional Min system such as WM4196, many FtsZ rings form aberrantly at cell poles and pinch off numerous minicells that are approximately the same diameter as the width of the mother cell. To further decrease the size of the minicells, we took advantage of a hypermorphic allele of the cell shape-determining actin homologue *mreB*. MreB controls cell width[26], and a MreB carrying the A125V substitution generates cells that are thinner than normal and has been used previously to make skinny minicell strains[27]. Therefore, we introduced the MreB-A125V substitution into WM4196 (see the "Methods" section)[28,29]. Finally, sequence comparisons with RP437, a wild-type reference strain for *E. coli* chemotaxis studies[30], identified a single base-pair mutation [*pflhDC(-10)g5a*] in WM4196 at the −10 region of the promoter for the *flhDC* genes, whose products promote expression of all flagellar and chemotaxis genes[31].

To ascertain the contributions of the *mreB*, *minD*, and *pflhDC* alleles to the favourable cryo-ET-imaging properties of WM4196 minicells, we constructed RP437 derivatives with combinations of the WM4196 alleles. In particular, we created an RP437 derivative (UU3118) carrying the *mreB-A125V* and *minD-G262D* alleles and isolated minicells from cultures of

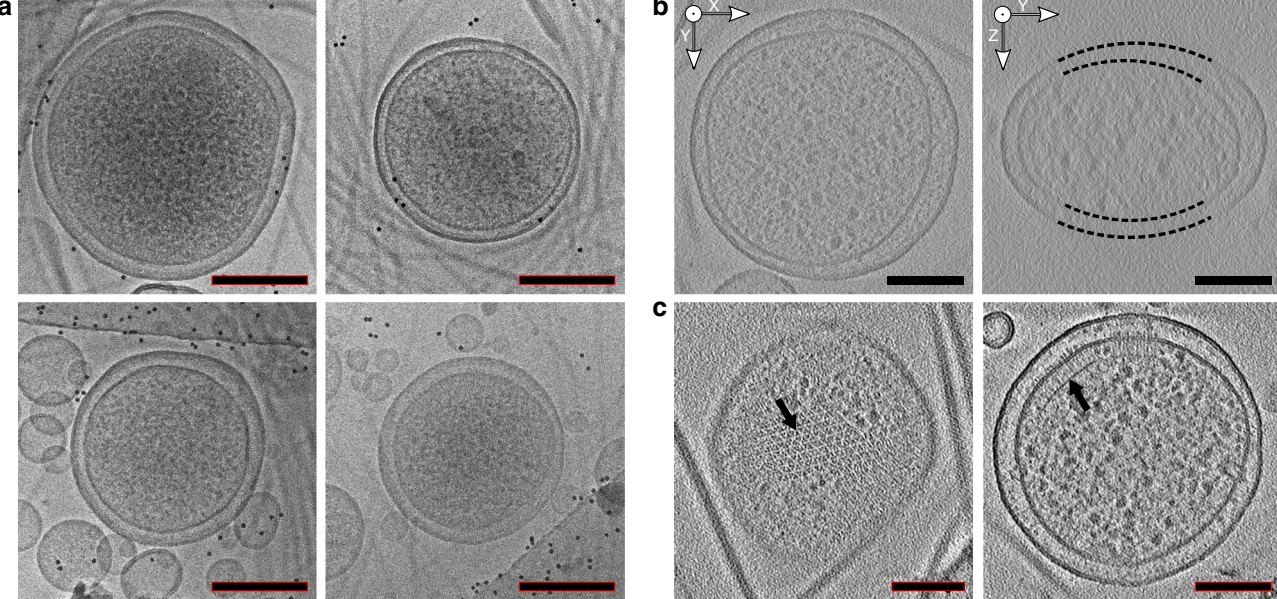

**Fig. 2 WM4196 minicells are suitable for high-resolution analysis of chemoreceptor arrays. a** Representative electron micrographs showing healthy looking WM4196 minicells. **b**—left XY slice and **b**—right YZ slices through tomograms of WM4196 minicells showing that they appear flattened in vitreous ice, yielding thin samples suitable for cryo-ET. Black dotted lines are used to show membrane positions which are not clear in the images due to missing-wedge effects from tomographic reconstruction. **c** slices through tomograms of WM4196 minicells exhibiting presence of chemoreceptor arrays aligned both perpendicular (left) and parallel (right) to the electron beam. Scale bar is 100 nm in all panels.

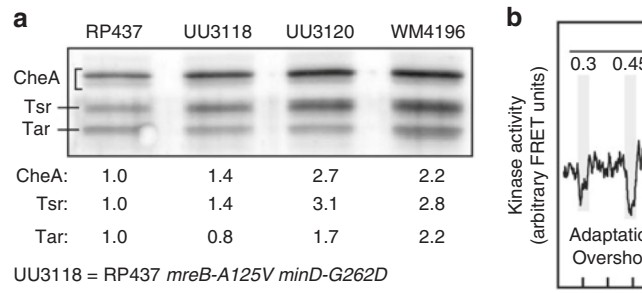
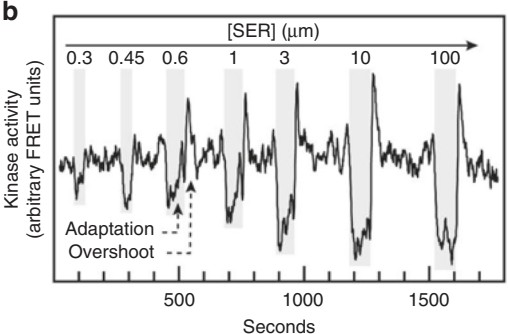
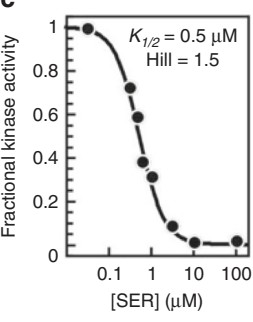

**Fig. 3 Chemotaxis proteins and signaling activities in WM4196 minicells. a** Tsr, Tar, and CheA levels in WM4196. Cell extracts were prepared and analysed by SDS–PAGE and western blotting as described in the "Methods" section. Band intensities were determined by densitometry and normalized to the RP437 values. The entire gel is shown in Supplementary Fig. 2. **b** FRET analysis of serine signaling in WM4196 minicells. WM4196 minicells expressing the CheY-YFP/CheZ-CFP FRET reporter pair (see the "Methods" section) were immobilized on polylysine-coated coverslips and mounted in a microscope flow chamber. The horizontal trace follows the ratio of YFP to CFP emission counts, a measure of CheA kinase activity in the cells. Serine stimuli were applied to the cells during the intervals marked by grey rectangles. The magnitude of the drop in YFP/CFP value reflects the fraction of CheA activity inhibited by a serine stimulus. Maximal CheA activity in the cells is defined by the YFP/CFP drop in response to a saturating serine concentration (e.g., 10 or 100 μM). These minicells contain the CheR and CheB adaptation enzymes which act to attenuate a serine-induced drop in kinase activity. A net methylation increase of the serine receptor during the adaptation phase produces a spike or overshoot in kinase activity upon serine removal that rapidly returns to the pre-stimulus kinase activity baseline[33]. **c** Hill fit of serine dose–response data. The fractional inhibition values from the experiment in panel **b** were fitted to a multi-site Hill function to determine the response $K_{1/2}$, a measure of receptor sensitivity, and the Hill coefficient, a measure of response cooperativity[50]. Source data are provided as a Source Data file.

WM4196 and UU3118 grown under identical conditions. The minicell sizes were compared by analysis of phase contrast light micrographs that showed that WM4196 minicells were overall smaller than those produced by UU3118 (Supplementary Fig. 1). Although we do not yet know the genetic basis for this size difference, a sequence comparison of the entire RP437 and WM4196 genomes is in progress and should identify candidate genes for subsequent study.

**The WM4196 strain possesses functional chemosensory array.** Our next aim was to characterise the WM4196 strain in terms of its suitability for structural analysis of the CSU. To this end, three complementary experiments were performed.

First, cultures of WM4196 and RP437 derivatives were examined for expression of three principal chemosensory array components: the Tsr and Tar chemoreceptors and the CheA kinase. Levels of all three components were 2–3-fold higher in WM4196 than in RP437 (Fig. 3a, Supplementary Fig. 2). Most of the expression increase likely derives from the *flhDC* promoter mutation, but the *mreB and minD* alleles combination of WM4196 also seems to contribute to the difference (Fig. 3a). An RP437 derivative (UU3120) carrying the *mreB*, *minD*, and *pflhDC* alleles of WM4196 produced comparably elevated expression of these array components, although the Tar:Tsr ratio was slightly different between the two strains (Fig. 3a).

Second, to determine whether the arrays in WM4196 cells support chemotaxis, we examined WM4196 colony morphology on semi-solid agar motility plates. Despite apparent up-regulation of chemotaxis and, presumably, flagellar genes, we found that WM4196 performed quite poorly in soft agar chemotaxis tests compared to RP437 derivatives UU3118 and UU3120 (Supplementary Fig. 3). The observed migration difficulties of WM4196 could be due to the fact that receptor arrays reside mainly at the poles of cells. Because WM4196 mother cells efficiently bud minicells at their poles, the minicells probably contain most of the receptor arrays, leaving the mother cells with relatively few. However, since basal bodies form at random locations around the cell[31], few minicells are likely to have both a receptor array and a flagellar motor. Regardless, any minicells capable of chemotaxis

would not contribute to colony expansion because they cannot reproduce. We surmise that the chemotaxis defects observed in RP437 are less severe than those observed in WM4196 (Supplementary Fig. 3) because the former bud minicells less efficiently and thereby have more array-containing mother cells during colony growth.

Third, to ask whether the receptor arrays in WM4196 minicells are capable of detecting and responding to attractant stimuli, we employed a FRET assay (see the "Methods" section) that monitors receptor-coupled CheA kinase activity in intact cells (Fig. 3b, c)[32,33]. WM4196 minicells detected serine with high sensitivity ($K_{1/2} = 0.4$ μM) and moderate cooperativity (Hill coefficient = 2.7), values comparable to RP437 responses (Fig. 3c). In addition, the minicell responses exhibited hallmarks of sensory adaptation (response decay and activity overshoot; Fig. 3b) consistent with a normal complement of the CheR and CheB receptor-modifying enzymes. The receptor arrays in WM4196 minicells thus function comparably to those in RP437 cells.

**3D cryo-ET map of the CSU shows a complete receptor density.** We next set out to calculate a 3D cryo-ET map of the CSU in WM4196 minicells (Fig. 4a, Supplementary Figs. 4–6, see the "Methods" section). In our cryo-ET reconstructions of individual minicells, the chemosensory array followed the curved surface of the inner membrane, providing the ensemble of views required for an isotropic 3D reconstruction by subtomogram averaging (Supplementary Fig. 4). Selection and iterative refinement of 681 particles from six tomograms in the Dynamo software package[34–36] resulted in a 3D density map of an extended array of CSUs. The final refinement was focused on a region containing a pseudo-hexameric arrangement of three transmembrane CSUs (Fig. 1c). According to the distribution of local-resolution estimates, the resolution of the resulting 3D reconstruction varies between ~15 and 30 Å (Supplementary Figs. 5 and 6, see also below). From this reconstruction of the higher-order array structure, one CSU was extracted for modelling and interpretation (Fig. 4a, Supplementary Fig. 4, see the "Methods" section).

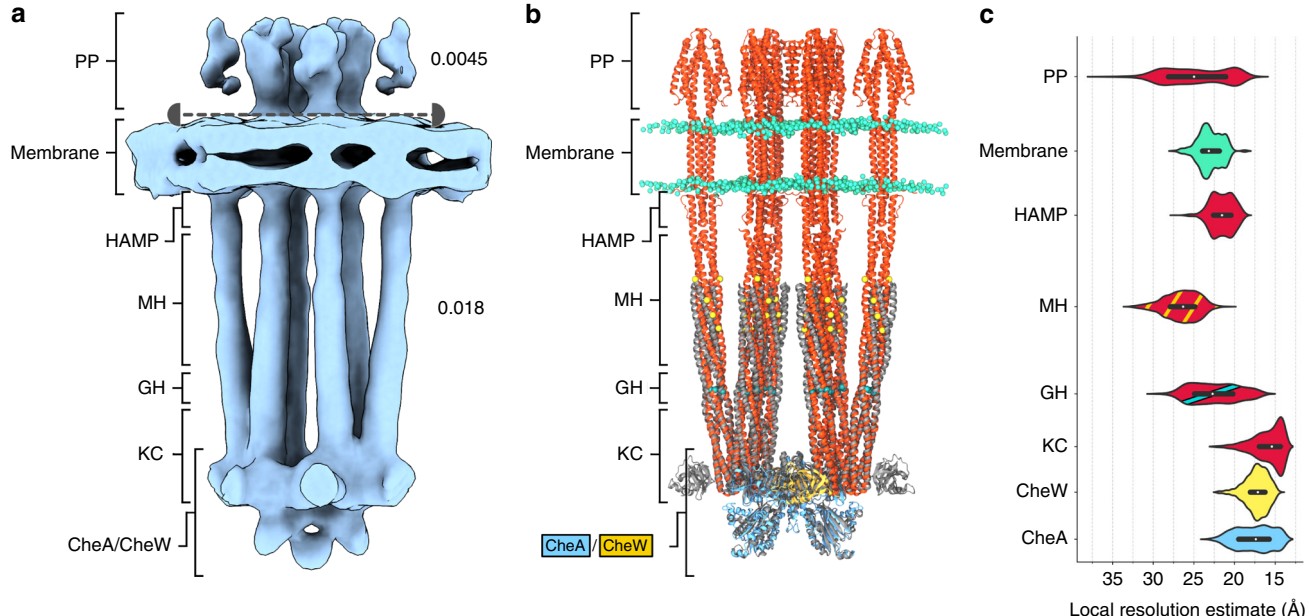

**Fig. 4 Cryo-ET map of the CSU interpreted in the light of the resulting molecular model. a** Subtomogram average of the CSU. The map is shown as an isosurface representation above and below the capped, dashed line at thresholds of 0.0045 and 0.018, respectively. Regions corresponding to the periplasmic ligand-binding domains (PP), the membrane, the HAMP domains, methylation-helix bundles (MH), flexible regions containing the glycine hinge (GH) and kinase control (KC) modules of the receptors are labelled, as well as the CheA/CheW baseplate. **b** Molecular model of the CSU. Receptors are depicted in red, CheA in blue, and CheW in gold. Membrane headgroups, methylation sites, and glycine hinge residues are shown as cyan, yellow, and teal spheres, respectively. The region of a previously published *T. maritima* CSU model (PDB 3JA6), modelled from a subtomogram average of *E. coli* chemoreceptor arrays, is superposed in grey. **c** Distribution of local resolution estimates in different regions of the CSU. Source data are provided as a Source Data file.

The most striking difference between our map and previously reported structures[9–13] is the complete and continuous receptor density, showcasing the entire assembly from the periplasmic ligand-binding domain to CheA and CheW at the cytoplasmic signalling tip (Figs. 1b and 4a). Indeed, densities corresponding to the periplasmic domain, transmembrane four-helix bundle, and HAMP domain were not resolved in previous array studies, which had been attributed to inherent flexibility of the CSU[2]. Although our current map probably derives from a mixture of receptors with a range of adaptational modifications and signalling states, both the HAMP and MH bundles seem to be relatively static. Thus, our 3D reconstruction demonstrates that complete CSUs are amenable to visualisation by cryo-ET. The WM4196 minicell system, therefore, represents a valuable tool for elucidating structure–function relationships in CSUs, for instance through the imaging of arrays in different mutationally imposed signalling states.

**Molecular model of the CSU and local resolution variations**. To gain further insight into CSU structure, we next sought to precisely localise the individual domains of each signalling component within our cryo-ET map. To this end, we constructed all-atom models of the *E. coli* CheA.P3.P4.P5 dimer, CheW monomer, and membrane-bound, full-length Tsr homodimer, making use of coordinates from existing high-resolution crystallographic structures where available (see the "Methods" section). We then assembled the component models according to the experimental density map and used Molecular Dynamics Flexible Fitting[37] to refine the tertiary protein structure as described previously[9,38].

The resulting molecular model (Fig. 4b) can be used to interpret local resolution distribution in each major region of the CSU cryo-ET map (Fig. 4c, Supplementary Figs. 5 and 6). Indeed, calculation of local resolution distributions shows large variations in local map quality and thus interpretability (Fig. 4c). The best-resolved regions of the map correspond to CheW and receptor signalling tips that contain the kinase control domain. The physical separation between this well-resolved cytoplasmic part of the structure and the periplasmic ligand-binding domains, combined with the relative flexibility of the MH bundle of the receptors, which is evidenced by the lower local resolution values in this region, could explain the lower map quality in the periplasmic space.

**Insights into transmembrane CSU structure and organisation**. Our molecular model (Figs. 4b and 5) of a complete transmembrane CSU provides several additional important structural insights. First, the model allows us to describe the previously uncharacterized periplasmic organisation of the receptors (Fig. 5b). In particular, the intact, membrane-bound MCP model, which was constructed separately from the map (see the "Methods" section), allows us to overcome local resolution differences in MCP density, enabling the positioning of the less-resolved MCP ligand-binding domains to be assisted by the better-resolved cytoplasmic domains during the fitting simulations. Surprisingly, we observe that in the periplasmic region the distances between receptors within a given ToD are similar to the inter-ToD distances (Fig. 5b, Supplementary Fig. 7), suggesting that minor diffusion within the membrane could give rise to ToD interactions both within and between CSUs. The impact of such interactions on signalling and cooperativity has yet to be systematically explored and should now be amenable to mutational and cross-linking analyses using WM4196-derived strains. In addition, the density-refined model demonstrates that the introduction of pronounced kinks in MCP structure, either between the HAMP and MH bundles or at the glycine hinge, is

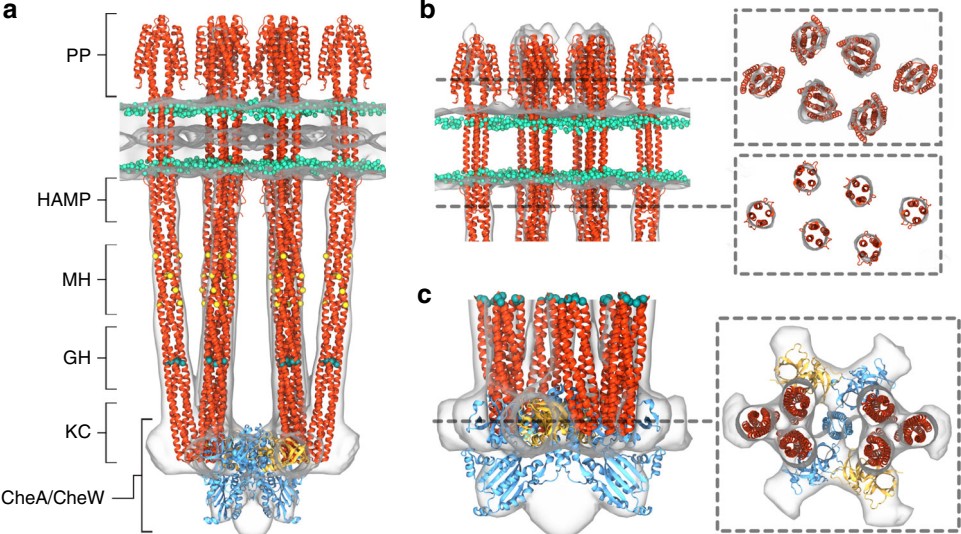

**Fig. 5 Molecular model of the complete *E. coli* core signalling unit. a** Overlay between CSU model and masked 3D density map shown with surface representation at an isovalue threshold of 0.016. Colours and labels in the model as in Fig. 4. **b** Zoom of membrane and membrane-proximal receptor domains from side (left) and at vertical slices taken through the periplasmic (top right) and HAMP (botton right) domains. To visualise the densities corresponding to the periplasmic domains, the map threshold is lowered to 0.0045 in the side view and the slice through the periplasmic domain. **c** Zoom of CheA/CheW baseplate region from side (left) and top (right). For the side view representation, the map and model have been rotated by 45° to highlight excess density between CheA.P4 domains. NB. Bulges of empty density at the level of the KC module, in both panel a and panel **c**, correspond to CheA or CheW molecules of neighbouring CSUs in the higher-order array structure not depicted in this visualisation. Source data are provided as a Source Data file.

not necessary to avoid structural clashes between neighbouring MCPs as previously suggested[39,40]. We note, however, that at the present resolution, the experimental map does not preclude the presence of such kinks in subsets of receptors or as transient occurrences, nor does it conflict with the idea that more gradual bending at the glycine hinge could play a role in transitions between signalling states. Overall, the symmetry axes of the ToDs are separated by 7.4 nm, suggesting an array lattice constant of ~12.8 nm which is consistent with inter-particle distances in the tomograms (Supplementary Fig. 8).

Examination of the CheA/CheW baseplate region in the present model further confirms the organisation deduced from earlier cryo-ET studies. In particular, the known MCP–CheA.P5, MCP–CheW, and CheA.P5–CheW interfaces critical for CSU assembly are all preserved (Fig. 5c). Moreover, direct comparison with a previously published cryo-ET-based model of the cytoplasmic portion of the *Thermotoga maritima* CSU (PDB 3JA6)[9] shows a nearly identical positioning of the CheA kinase (Supplementary Fig. 9): the CheA.P3 dimer sits between and parallel to the two ToDs while the CheA.P4 domain is localised in a similar orientation directly beneath CheA.P5 (Fig. 5c). Intriguingly, the refined model draws attention to a substantial amount of density situated below and between the two CheA.P4 domains that is not accounted for (Fig. 5c). Several lines of evidence suggest that the CheA.P1 and CheA.P2 undergo dynamic changes during signalling, and these domains have been proposed to be sequestered in the kinase-OFF state[5,12]. In addition, NMR experiments have mapped a non-productive P1–P4 binding mode to this region of CheA.P4 in *T. maritima*[41]. Therefore, the unknown density may correspond to either one or both of the CheA.P1 domains and could also include CheA.P2. In general, the role of dynamics in CheA signalling and the mechanisms of receptor regulation are matters of debate[4]. Thus, our WM4196 minicell system provides a much-needed tool for assessing these issues experimentally.

## Discussion

While the primary aim of this work was to solve the complete structure of a single CSU, our approach may be extended to the higher-order array structure. This would enable an investigation of the molecular mechanisms underlying long-range and cooperative signal transduction between CSUs. One example would be the analysis of array packing and dynamics in different signaling states. The WM4196 strain thus also represents a useful tool for investigating the molecular mechanisms underlying the remarkable signal amplification and wide dynamic range of the chemosensory system[14,42].

In summary, owing to their size and structural integrity, the WM4196 minicells presented in this work reveal themselves as suitable for cryo-ET studies. They are also particularly appropriate for higher resolution analysis of chemosensory arrays due to their elevated expression level of chemosensory components compared to the RP437 reference strain. Moreover, because WM4196 minicells bud closer to the bacterial pole, they contain a higher percentage of arrays than larger minicell strains, enabling more efficient acquisition of usable tilt-series for subsequent subtomogram averaging. Here we have shown that properties of the WM4196 strain lead to a significant improvement in the quality of the in situ CSU structure with an unparalleled view of the MCP structure, including the periplasmic ligand binding, HAMP, and the MH bundle domains that play a vital role in converting ligand-induced conformational rearrangements into CheA control (Supplementary Movie 1). The results obtained in the present work can now be fruitfully combined with higher resolution studies of in vitro reconstituted chemosensory arrays[9] and compared with cryo-ET reconstructions from other bacteria, such as *Vibrio cholerae*, *Rhodobacter sphaeroides* and *Caulobacter crescentus*[43–46]. Finally, these small and healthy minicells open up opportunities for the study of *E. coli* macromolecular complexes located near the cell poles.

## Methods

**Design and genetic characterisation of the WM4196 strain**. Strain WM4196 is a distant derivative of a minicell mutant first described over 50 years ago. This prototype strain, containing the *minD* allele, a *dnaB* ts allele, as well as several auxotrophic markers, was converted to *dnaB+* and prototrophy by conjugation with a prototrophic Hfr donor strain[29]. The F′ was then cured to make strain x1411, which was obtained from the *E. coli* Genetic Stock Center, Yale University (CGSC#6397). The *mreB*-A125V allele[28] was further subsequently introduced into x1411 by cotransduction with a tightly linked *yhdE::cat* chromosomal allele to create the WM4196 strain.

**DNA sequencing and strain constructions**. WM4196 chromosomal DNA was PCR-amplified and sequenced with primer pairs specific for the *minCDE*, *flgM*, and *flhDC* loci. Strain UU3118 (RP437 *mreB-A125V minD-G262D*) was constructed by (a) introducing the *mreB* allele from WM4196 into RP437 by phage P1-mediated cotransduction with the linked *ΔyhdE::cat* allele; and (b) introducing the *minD-G262D* mutation with homologous recombination-mediated insertion and replacement steps[47]. Strain UU3120 is a derivative of UU3118 that carries the *pflhDC (-10)g5a* allele of WM4196 introduced by two-step insertion/replacement.

**Measuring cellular levels of chemotaxis proteins**. Plasmid pRR48[48] was introduced into RP437, WM4196, and UU3111 to express the β-lactamase (Bla) protein (an internal reference for standardizing the relative levels of other cell proteins). Cells were grown at 37 °C in L broth (10 g l$^{-1}$ tryptone, 5 g l$^{-1}$ yeast extract, 5 g l$^{-1}$ NaCl) containing 25 μg ml$^{-1}$ ampicillin and harvested at mid-exponential phase (OD$_{600}$ = 0.5–0.6). Approximately equal numbers of cells (based on culture OD) were used to prepare lysates. Cells were pelleted by centrifugation, washed once with motility buffer (10 mM K-PO$_4$ [pH 7.0], 0.1 mM K-EDTA) and lysed in SDS–PAGE buffer[49]. Samples were analysed in 11% polyacrylamide–SDS gels and MCP, CheA and Bla bands were detected by immunoblotting with a mix of three polyclonal rabbit antisera. Bands were visualized with a Cy5-labeled goat anti-rabbit antibody and quantified with a fluorescence imager. The uncropped and unprocessed scan of the gel is shown in Supplementary Fig. 2 and in the Source Data file to Fig. 3a.

**In vivo FRET kinase assay**. This method is described in detail in previous publications[32,50]. For the present study, we introduced into WM4196 plasmid pVS88[33], which expresses fusion proteins CheZ-CFP (FRET donor) and CheY-YFP (FRET acceptor) under IPTG-inducible control. The plasmid-carrying strain was grown at 37 °C to OD$_{600}$ = 0.65–0.7 in L broth (see above) containing 25 μg ml$^{-1}$ ampicillin and 1 mM IPTG. (The high IPTG concentration compensates for elevated levels of untagged CheY and CheZ proteins expressed from the WM4196 chromosomal genes.) 300 ml of cell culture were centrifuged at ~8600 × *g* for 25 min at 4 °C. All subsequent centrifugation steps were also carried out at 4 °C. The supernatant was transferred to fresh tubes and centrifuged at ~14,500 × *g* for 25 min. The sample pellets were pooled and resuspended in ~1 ml of motility buffer (see above) and carried through a second round of differential centrifugation (9000 × *g* for 10 min; 21,000 × *g* for 20 min). The final minicell pellet was resuspended in ~100 μl motility buffer and applied to a polylysine-coated cover slip for the FRET assay.

**WM4196 minicell purification for cryo-ET imaging**. WM4196 minicells were grown at 37 °C in L broth supplemented with 34 μg ml$^{-1}$ chloramphenicol for 12 h. Small volumes of this culture were used to inoculate larger volumes of L broth media (without antibiotics) to an initial OD$_{600}$ value of 0.075. These larger volume cultures were grown at 37 °C for 4 h, the final OD$_{600}$ value was 1.75. The culture was centrifuged at 8683 × *g* for 20 min at 4 °C, the supernatant carefully transferred to another centrifuge tube and centrifuged again at 8683 × *g* for 20 min at 4 °C. The resulting supernatant was transferred to another centrifuge tube and centrifuged at 41,500 × *g* for 20 min at 4 °C. This time, the supernatant was discarded and the pellet gently resuspended in the residual supernatant from the centrifuge tube. This suspension was then centrifuged at 5500 × *g* for 5 min at 4 °C, and the supernatant transferred to another centrifuge tube to be centrifuged at 16,900 × *g* for 15 min at 4 °C. The resulting minicell pellet was resuspended in LB and the minicell suspension was placed at 4 °C.

**Cryo-EM imaging and cryo-ET data acquisition**. Minicell solution was supplemented with 10 nm colloidal protein A-gold particles (Cell Microscopy Core, University Medical Center, Utrecht, The Netherlands). R 2/1 or R 3.5/1 on 300 mesh Cu/Rh grids (QUANTIFOIL) were glow discharged for 45 s at 25 mA. 3 μl of the sample was applied to grids and plunge frozen in liquid ethane using a Vitrobot Mark IV (Thermo-Fisher Scientific). Grids were stored under liquid nitrogen until imaging. Screening of the grids was performed on an FEI F20 microscope at 200 keV. Tilt-series acquisition was performed on a Titan Krios transmission electron microscope operated at 300 keV, through a Gatan Quantum energy filter and Volta phase plate onto a K2 Summit direct electron detector, using SerialEM software[51]. For both search and navigational purposes, low-magnification montages were acquired. For tilt-series acquisition, a magnification which produced a calibrated pixel size of 2.24 Å was selected. Data were collected at tilt-angles

between −60° and +60° in 2° increments in a grouped dose-symmetric tilt scheme (0, +2, +4, +6, +8, +10, −2, −4, −6, −8, −10, 12, 14, 16, 18, 20, −12, −14, −16, −18, −20, etc.)[52]. Images at each tilt-angle were acquired as movies comprising five frames. Target total dose for tilt-series acquisition was 60 e$^-$ Å$^{-2}$.

**Image processing and pre-processing**. Frames of movies corresponding to images at each tilt-angle in each tilt-series were aligned and motion-corrected using MotionCor2[53] to mitigate the deleterious effects of beam-induced sample motion. Tilt-series were assembled using the newstack command from IMOD[54].

**Tomographic reconstruction**. Tilt-series were aligned, binned by two and tomograms were reconstructed by weighted back-projection using the IMOD software package. Six tomograms of WM4196 minicells were selected for further processing.

**Initial reference generation**. Particles were identified and picked using the Dynamo software package[34–36]. Sub-volumes with a side length of 573 Å were extracted and initial orientations for each particle were estimated by modelling a surface following the curvature of the chemosensory array in the tomogram and imparting an orientation onto each particle, which corresponded to the normal to the modelled surface at the point closest to the particle. These orientations were further refined manually, and these coarsely oriented particles were then averaged to produce an initial reference. The particles were subsequently locally aligned, constraining the angular search to a 60° cone around the initial estimate for the orientation, and averaged to produce initial references.

**Particle picking**. The following analysis was performed in Dynamo. Surfaces were modelled following the curvature of the chemoreceptor arrays visible inside the minicells, and a set of initial positions and orientations was generated from this surface with an average distance of 30 Å between each position. Sub-volumes with a side length of 573 Å were extracted at each of these positions and each was aligned to the initial reference with an allowed translational freedom of 60 Å in each of the *x*-, *y*- and *z*-dimensions. Analysis of the post-alignment positions revealed an ordered hexagonal array. Duplicate particles (defined as particles within 10 Å of each other) were collapsed into one final position. The particle positions were then cleaned by selecting only those which had greater than three nearest-neighbours at the expected distance of 120 ± 20 Å.

**Subtomogram averaging**. Alignment and averaging of the particles was performed in the Dynamo software package. An iterative global alignment of the particle positions and orientations was performed starting from the initial reference already generated, for which alignments were performed inside a mask encompassing multiple CSUs in the chemosensory array and the inner membrane of the WM4196(DE3) minicell. These positions and orientations were then further locally refined inside a mask containing only three CSUs, without the membrane[15], to produce a final reconstruction resulting from 681 subtomograms derived from six tomograms.

**Post processing**. Two separate half-maps were generated from groups of particles coming from different tomograms to allow estimation of the resolution of the final reconstruction. The Fourier shell correlation (FSC) of these maps inside a mask, containing three CSUs and no membrane, drops below 0.5 and 0.143 at spatial frequencies corresponding to resolution estimates of 18 and 16 Å, respectively (Supplementary Fig. 5). The maps were subsequently subjected to localised resolution estimation in RELION[55] with a sampling rate of 20 Å and locally filtered to the estimated resolution of the map. A histogram of the per-voxel local resolution estimations for regions inside the FSC mask shows that most areas of the map fall in the range 15–25 Å (Supplementary Figs. 5 and 6, Fig. 4c). The local-resolution-filtered map was aligned to a C2-symmetric reference centred on one CSU and then itself symmetrised around the C2-axis to give a final map centred on one CSU. Separate masks for FSC calculation and map visualisation were calculated using a combination of Chimera[56], Dynamo and RELION.

**Molecular modelling and simulations**. Atomic coordinates for *E. coli* CheA.P3. P4.P5 and CheW were derived from existing *T. maritima* crystal structures[57,58] using template-based homology modelling. An all-atom model of the *E. coli* Tsr (residues 1–518) was constructed from existing crystal structures of the ligand binding and cytoplasmic domains as well as a HAMP homology model based on a crystal structure of *A. fulgidus* HAMP. The transmembrane four-helix bundle was modelled using existing cross-linking data between the individual helices. As the long, unstructured C-terminus of Tsr (residues 519–551) was not resolved in the cryo-ET map, we did not include these residues in the model. The intact Tsr model was then embedded in a 3:1 POPE:POPG lipid bilayer and equilibrated for 500 ns using molecular dynamics simulation, providing an initial structure for further conformational refinement via MDFF.

All-atom molecular dynamics simulations were carried out using NAMD 2.13[59] (and the CHARMM36 force field[60]). MDFF simulations were performed in the NVT ensemble at 310 K in explicit solvent. A scaling factor of 0.1 was used to couple backbone atoms to the MDFF potential. To prevent the loss of secondary structure as

well as the formation of cis-peptide bonds and chirality errors, additional harmonic restraints were applied to the protein backbone during the fitting simulations.

**Reporting summary**. Further information on research design is available in the Nature Research Reporting Summary linked to this article.

## Data availability

Data supporting the findings of this manuscript are available from the corresponding author upon reasonable request. A reporting summary for this Article is available as a Supplementary Information file. The Cryo-ET derived map has been submitted to the EMDB with accession code EMD-10160. The source data underlying Figs. 1b, c, 3a, b, 4c, 5 and Supplementary Figs. 1, 5a, c and 8 are provided as a Source Data file.

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

## Acknowledgements

This work has received funding from a European Union's Horizon 2020 research and innovation programme under grant agreement No. 647784 to I.G. Research in the JSP and WM labs was supported by grants GM19559 and GM131705, respectively, from the U.S. National Institute of General Medical Sciences. This work was also supported by the U.S. National Institutes of Health grant P41GM104601 to C.K.C. and Z.L.-S., the U.S. National Science Foundation grant PHY1430124 to C.K.C. and Z.L.-S., and the U.K. Biotechnology and Biological Sciences Research Council grant BB/S003339/1 to C.K.C. and P.J.S. P.J.S.'s lab is also supported by Wellcome (208361/Z/17/Z), the BBSRC (BB/P01948X/1, BB/R002517/1) and the MRC (MR/S009213/1). The Project made use of the ARCHER UK National Supercomputing Service (http://www.archer.ac.uk), provided by HECBioSim, the UK High End Computing Consortium for Biomolecular Simulation (hecbiosim.ac.uk), which is supported by the EPSRC (EP/L000253/1). We acknowledge Diamond Light Source for access and support of the cryo-EM facilities at the UK's national Electron Bio-imaging Centre (eBIC), funded by the Wellcome Trust, MRC and BBRSC. Cryo-ET data acquisition has been supported by iNEXT, grant number 653706 (PID:2626 to I.G.), funded by the EU Horizon 2020 programme. For initial minicell characterisation and grid screening, we used the platforms of the Grenoble Instruct-ERIC Center (ISBG: UMS 3518 CNRS-CEA-UGA-EMBL) with support from FRISBI (ANR-10-INSB-05-02) and GRAL, a project of the University Grenoble Alpes graduate school (Ecoles Universitaires de Recherche) CBH-EUR-GS (ANR-17-EURE-0003). IBS acknowledges integration into the Interdisciplinary Research Institute of Grenoble (IRIG, CEA). The IBS electron microscope facility is supported by the Rhône-Alpes Region, the Fondation pour la Recherche Médicale (FRM), the fonds FEDER, the Centre National de la Recherche Scientifique (CNRS), the Commissariat à l'Energie Atomique et aux Energies Alternatives (CEA), the University of Grenoble Alpes, EMBL, and the GIS-Infrastructures en Biologie Santé et Agronomie (IBISA). Molecular dynamics simulations were performed on the Blue Waters supercomputer, which is supported by the National Science Foundation (OCI-0725070 and ACI-1238993) and the state of Illinois as part of the Petascale Computational Resource Grant (ACI-1713784). We are particularly grateful to Daniel Clare, Alistair Siebert and Andrew Howe for help with data acquisition at eBIC, and to Daniel Castaño-Diez for help and discussions on image processing and for development of Dynamo. We thank Guy Schoehn for establishing and managing the IBS Grenoble cryo-electron microscopy platform and for providing training and support. We are grateful to Aymeric Peuch for help with the usage of the joint IBS/EMBL Grenoble EM computing cluster.

## Author contributions

I.G. designed and supervised the study. W.M. created the WM4196 minicell strain. A.B., M.B. and K.H. purified the minicells, M.B.-V., A.B. and I.G. screened them by cryo-EM. P.A., W.M. and J.S.P. created the RP437 derivatives, and characterised them and the WM4196 strain. A.B., A.D. and I.G. collected cryo-ET data, A.B. analysed cryo-ET data with input from A.D. and I.G. C.K.C. constructed and refined the molecular model in discussion with Z.L.-S. and P.J.S. and interpreted it in the context of the cryo-ET density with input from A.B. and I.G. A.B., C.K.C., P.A., J.S.P. and W.M. prepared the figures and movies. I.G. wrote the manuscript with significant input from A.B., C.K.C., P.J.S., W.M. and J.S.P. and contribution of all authors.

## Competing interests

The authors declare no competing interests.
