## [Peer Review File · Nature Communications]

Reviewers' Comments:

Reviewer #1:

Remarks to the Author:

Over the last decade cryo-ET has significantly enhanced our understanding of chemosensory arrays and signaling complexes and there is no doubt that this rapidly evolving technique will be a central tool to study complex protein assemblies in the future. The authors report here the generation of an optimized *E. coli* minicell that, due to its reduced size, is better suited for cryo-ET studies than "normal" *E. coli* cells. Although swim plates show that this strain was somewhat impaired in chemotaxis, FRET measurements revealed responses to serine with a EC50 value of 0.4 micromolar; a value close to the serine dissociation constant for its receptor, which indicates effective signal transduction.

The authors report a 16 Å structure of the entire receptor. The major advance of this structure is the clear density for the chemoreceptor sensor domain that was missing in previous structures. The observation that sensor domains from different trimers of dimers are in closer distance than within the trimer of dimer will without any doubt motivate research to explore the relevance of sensor domain contacts in chemoreceptor arrays. So far, individual four helix bundle domains were found to be present in a monomer-dimer equilibrium, but there is evidence for small amounts of higher order oligomeric states.

At line 54 the authors state "The mechanistic details of sensory signal transmission within the CSU remain mysterious...". In a scientific publication it is important to highlight the novelty and potential impact of the data reported, but this sentence is an over-simplification, considering the enormous amount of publications on the signal transmission within a CSU. The authors should do a better job and state what is known, what is unknown and then make the case that this new approach is a convenient means to fill the gaps of knowledge identified.

Tar and Tsr possess a terminal pentapeptide that corresponds to the high-affinity binding site for CheR. The inspection of Fig. 1C shows density at approximately the place of where one would expect this pentapeptide. This referee is aware that in "normal" *E. coli* CheR is not very abundant, but could this density correspond to bound CheR? Have measurements been done with chemoreceptors that lack the terminal pentapeptide?

There are also two well-defined bulges of density to both sides of the tip (Fig. 4A)? Do the authors have any idea on what this may be?

Reviewer #2:

Remarks to the Author:

This manuscript presents an important advance in the study of bacterial chemosensory arrays. The newly developed mini-cell producing *E. coli* strain described herein allows for cryo-tomographic imaging in samples that produce arrays that are more ordered than previously observed and hence can be visualized completely. For the first time, the entire chemoreceptor can be positioned in the array. The resulting models reveal periplasmic domain arrangements and receptor conformations not before seen. The approach offers much promise for further study. I find the paper very readable and the data easy for the reader to interpret. I only have several comments for the authors:

1. I'm uncomfortable with the statement of lines 167-169 that the glycine hinge and kink at the HAMP domain are "clearly not required" to avoid structural clashes. When I look at the density, it does appear to me that the receptor is kinked below the HAMP and somewhat around the Gly hinge. Although the models are fit reasonably with a more continuous bend, it may be that a model with two slightly more pronounced kinks in these regions would fit these still relatively low-resolution maps as well. Greater quantification on the range of receptor conformations in agreement with the averages would help support the statement.

2. Lines 87-90. It would be helpful for the general reader to have a few sentences on the roles of

mreB and minD in cell structure and division, as well as why these mutations may produce beneficial mini-cell properties.

3. line 312+ - how many subtomograms were averaged for the final map? Supplemental Fig. 3 is difficult to read – the axis labels for example are too small.

4. A note on the number of subtomograms per array (i.e. array size) in the minicells would also be interesting in comparison to studies of other sample types.

5. From the histogram of Supp. Fig. 1, it's not clear to me that the size distribution of WM4196 is actually tighter than that of UU3118 (it does certainly have a lower mean). WM4196 looks more Gaussian, but it also seems to have a higher variance.

Reviewer #3:

Remarks to the Author:

The manuscript by Burt, Cassidy et al report a new minicell line for studying bacterial chemoreceptor arrays and perform subtomogram averaging to show an atomistic model of the system. I think there are several issues with the subtomogram averaging, which I would need a clarification from the authors before I am convinced about the structure. I am also not sure it is appropriate to call this an "atomistic structure" with such low-resolution. While I think the authors could be given a chance to improve their manuscript in a revised version, it would have to be significantly improved to convince this reviewer.

1. Generally speaking, I think the lack of an FSC curve, threshold levels, local resolution estimation plotted onto the structure, particle numbers, make it very difficult to interpret the quality of their map. Could the authors provide these?

2. The map is already available on EMDB and I looked at it. The periplasmic domain is only visible at a very low contour level. Worryingly, only a few periplasmic domains are visible, while the long alpha helical bundles are much stronger in density and I see more of them. Should the numbers of the long alpha helical units and the periplasmic domain not be the same? This means that the periplasmic domain is disordered. I would like to see the structure of this domain validated somehow.

- perhaps you could you provide local resolution estimate for this part?

- could the authors report a goodness of fit to a previous atomic structure?

3. The map does not look like a 16 Å resolution map to me. FSC curve would be helpful. Reading the methods section, it seems as if all subtomograms were aligned together but averaged in two halves for the FSC measurement. In this case, please use the 0.5 criterion for resolution estimation. You can use 0.143 if the maps have been aligned and averaged completely independently (see Scheres and Chen, Nat Methods).

4. Concerning the new minicell line, I cannot see the loading controls in the gel although the authors claim to use one in the methods section. Could you show these please?

5. Additionally one main point of the manuscript is that their new cell line is a useful tool to study chemosensory arrays because those minicells are very small and have to 50% those arrays. What is "n" here? How many cells were imaged? I am unconvinced that this is a huge improvement. The authors should try and focus on the biology revealed by the structure.

6. Additionally there are quite a few typos or inconsistencies in the format between the sections. For example "chemoecetpor" in Line591, switching between ml and mL or "x g" and "g", a total dose as e- per Å² per sec instead of e- per Å². Could the authors please standardise the

manuscript.

Reviewer #4:

Remarks to the Author:

This is a potentially landmark study. However, I have a lot of concerns with the brevity of the all-atoms model. I do not think it is 'wrong', it appears to be a nice work actually, from a technical point of view, but I do not see its point either, because there is no discussion of the model at all or comparison of the model with previous published models of the (partial, admittedly) chemotaxis apparatus models.

I understand that the all-atoms model is 'to be published later' in details, but that statement is not enough in my opinion to justify the publication of the model. Could the authors at least comment on whether or not their model is compatible with previous published dynamics mechanism of chemotaxis, some of them, apparently, with co-authorship by some of the author(s) of the present manuscript? For instance:

<https://www.ncbi.nlm.nih.gov/pmc/articles/PMC3123975/>

<https://elifesciences.org/articles/08419>

<https://journals.plos.org/ploscompbiol/article?id=10.1371/journal.pcbi.1002685>

http://chemotaxis.biology.utah.edu/Parkinson_Lab/publications/PDFs/Ortega%20et%20al.,%202013.pdf

Point-by-point response to reviewers' comments

Reviewer #1 (Remarks to the Author):

Over the last decade cryo-ET has significantly enhanced our understanding of chemosensory arrays and signaling complexes and there is no doubt that this rapidly evolving technique will be a central tool to study complex protein assemblies in the future. The authors report here the generation of an optimized *E. coli* minicell that, due to its reduced size, is better suited for cryo-ET studies than “normal” *E. coli* cells. Although swim plates show that this strain was somewhat impaired in chemotaxis, FRET measurements revealed responses to serine with a EC₅₀ value of 0.4 micromolar; a value close to the serine dissociation constant for its receptor, which indicates effective signal transduction.

The authors report a 16 Å structure of the entire receptor. The major advance of this structure is the clear density for the chemoreceptor sensor domain that was missing in previous structures. The observation that sensor domains from different trimers of dimers are in closer distance than within the trimer of dimer will without any doubt motivate research to explore the relevance of sensor domain contacts in chemoreceptor arrays. So far, individual four helix bundle domains were found to be present in a monomer-dimer equilibrium, but there is evidence for small amounts of higher order oligomeric states.

1. At line 54 the authors state “The mechanistic details of sensory signal transmission within the CSU remain mysterious...”. In a scientific publication it is important to highlight the novelty and potential impact of the data reported, but this sentence is an over-simplification, considering the enormous amount of publications on the signal transmission within a CSU. The authors should do a better job and state what is known, what is unknown and then make the case that this new approach is a convenient means to fill the gaps of knowledge identified.

We have considerably altered the introductory discussion (lines 65-69) to better highlight the substantial amount of previous work done to characterize signalling within MCPs, CheA, and the larger array, adding references to several key reviews. We also now emphasize that what is missing is the synthesis and expansion of previous work into a comprehensive, residue-level picture of signalling within the CSU complex, and especially between the constituent proteins.

2. Tar and Tsr possess a terminal pentapeptide that corresponds to the high-affinity binding site for CheR. The inspection of Fig. 1C shows density at approximately the place of where one would expect this pentapeptide. This referee is aware that in “normal” *E. coli* CheR is not very abundant, but could this density correspond to bound CheR? Have measurements been done with chemoreceptors that lack the terminal pentapeptide? There are also two well-defined bulges of density to both sides of the tip (Fig. 4A)? Do the authors have any idea on what this may be?

The terminal pentapeptide referred to by the reviewer is connected to the MCP cytoplasmic four-helix bundle by a long, unstructured segment roughly 30 residues long (see, for example, Fig. 1 from Parkinson et al., 2015, Trends in Microbiology). It's unlikely that such a flexible linker would be resolved in the cryo-ET maps. Given the relatively low resolution of the reconstruction in these areas, we prefer not to attempt interpretation of small bulges of density. Indeed, it is not present in the averaged, 3D maps of the CSU. As such, this portion of the receptor (residues 519-551) was not included in the Tsr model. To prevent confusion, we now state explicitly in the Methods section which Tsr residues were modelled (lines 411-413) and refer the reader to this section in the main text discussion of the Tsr model. Regarding the bulges of empty density at the level of the KC module, we now state (legend to Figure 5) that they correspond to CheA or CheW molecules of neighbouring CSUs in the higher-order array structure from which one CSU was extracted for the analysis.

Reviewer #2 (Remarks to the Author):

This manuscript presents an important advance in the study of bacterial chemosensory arrays. The newly developed mini-cell producing *E. coli* strain described herein allows for cryo-tomographic imaging in samples that produce arrays that are more ordered than previously observed and hence can be visualized completely. For the first time, the entire chemoreceptor can be positioned in the array. The resulting models reveal periplasmic domain arrangements and receptor conformations not before seen. The approach offers much promise for further study. I find the paper very readable and the data easy for the reader to interpret. I only have several comments for the authors:

1. I'm uncomfortable with the statement of lines 167-169 that the glycine hinge and kink at the HAMP domain are “clearly not required” to avoid structural clashes. When I look at the density, it does appear to me

that the receptor is kinked below the HAMP and somewhat around the Gly hinge. Although the models are fit reasonably with a more continuous bend, it may be that a model with two slightly more pronounced kinks in these regions would fit these still relatively low-resolution maps as well. Greater quantification on the range of receptor conformations in agreement with the averages would help support the statement.

Our intention in the cited lines was to highlight that it is not necessary to introduce kinks at any place along the MCPs in order to obtain clash-free structures of the CSU and extended array. It was previously suggested, based on negative-stain EM and geometric arguments (Akkaladevi et al., 2018, J. Bacteriol.; Stalla et al., 2019, Int. J. Mol. Sci.), that kinks between the HAMP and MH bundles and at the glycine hinge would be required to allow MCPs to pack into the known array configuration. However, we show that gradual MCP bending would be sufficient. Nevertheless, we did not mean to suggest that kinks are certainly not present in any receptors at any time. Thus, we now provide several caveats to our observation to better account for limitations in the interpretation of our cryo-ET data, namely that kinks could potentially exist within subsets of receptors or in a transient or less-pronounced way during signalling. These issues have now been clarified in the main text (lines 230-236).

2. Lines 87-90. It would be helpful for the general reader to have a few sentences on the roles of mreB and minD in cell structure and division, as well as why these mutations may produce beneficial mini-cell properties.

We now add descriptions of the roles of MreB and MinD in lines 111-119.

3. line 312+ - how many subtomograms were averaged for the final map? Supplemental Fig. 3 is difficult to read – the axis labels for example are too small.

In lines 179-181 we now specify that we used 681 particles from 6 tomograms. We made Supplementary Figure 3 bigger and put the text in the legend rather than in the figure itself to facilitate the viewing of the figure. This figure is meant only to serve illustrative purposes, while the exact image analysis strategy is presented in the Methods section.

4. A note on the number of subtomograms per array (i.e. array size) in the minicells would also be interesting in comparison to studies of other sample types.

The size of the array is actually quite variable as illustrated in this figure:

5. From the histogram of Supp. Fig. 1, it's not clear to me that the size distribution of WM4196 is actually tighter than that of UU3118 (it does certainly have a lower mean). WM4196 looks more Gaussian, but it also seems to have a higher variance.

We now rephrase this sentence (lines 132-134) and provide a different, more clear representation, of the frequency distribution of minicell areas in Supplementary Fig. 1.

Reviewer #3 (Remarks to the Author):

The manuscript by Burt, Cassidy et al report a new minicell line for studying bacterial chemoreceptor arrays and perform subtomogram averaging to show an atomistic model of the system. I think there are several issues with the subtomogram averaging, which I would need a clarification from the authors before I am convinced about the structure. I am also not sure it is appropriate to call this an "atomistic structure" with such low-resolution. While I think the authors could be given a chance to improve their manuscript in a revised version, it would have to be significantly improved to convince this reviewer.

We would like to specify that "atomistic" does not mean "atomic". However, we don't imply equal confidence in the positions of all atoms, and we have now removed the word "atomistic" in reference to the molecular model throughout the manuscript. The term "all-atom" has been kept in several places because it is necessary to differentiate the present models and simulations from coarse-grained simulations, which has important consequences for their accuracy. We have also taken care not to refer to the coordinates as a "structure" but rather as a model. In addition, we now state explicitly in the main text (lines 205-207) that only the protein tertiary structure has been refined by the cryo-ET data.

1. Generally speaking, I think the lack of an FSC curve, threshold levels, local resolution estimation plotted onto the structure, particle numbers, make it very difficult to interpret the quality of their map. Could the authors provide these?

While this information is regularly provided for single particle image analysis, it is yet far from being the case in the field of cryo-ET and the majority of published papers don't present the FSC curve or the local resolution estimation for 3D maps obtained by subtomogram averaging. We agree that this information should be available to the reader and now provide it in Supplementary Fig. 4.

2. The map is already available on EMDB and I looked at it.

Indeed, we would like to emphasize that although more than a dozen cryo-ET maps of the CSU in situ have been published, we are the only ones who have deposited our map in EMDB to make it accessible to the entire cryo-ET and chemoreceptor array communities. In fact, the only previously deposited cryo-ET map is the one of a reconstituted array composed of receptor cytoplasmic domains, CheA and CheW that are assembled on lipid monolayers (Cassidy et al, 2015, eLife). It is true that in the absence of deposited structures, one is limited to a visual comparison of our map with the maps depicted in the figures of the previous publications. This figure illustrates the difference between our map and the currently best published map, visible in particular at the level of the complete and continuous receptor density that, in the case of our map, showcases the entire assembly from the periplasmic ligand-binding domain to CheA and CheW at the cytoplasmic signalling tip. The completeness of the structure is, in our view, even more important than the nominal resolution (see also below).

The periplasmic domain is only visible at a very low contour level. Worryingly, only a few periplasmic domains are visible, while the long alpha helical bundles are much stronger in density and I see more of them. Should the numbers of the long alpha helical units and the periplasmic domain not be the same? This means that the periplasmic domain is disordered. I would like to see the structure of this domain validated somehow.

- perhaps you could you provide local resolution estimate for this part?

We now provide the threshold values and the local resolution distributions for each domain of the 3D map at Fig. 4 and Supplementary Fig. 4 (see details further below).

- could the authors report a goodness of fit to a previous atomic structure?

We have now included in Fig. 4 and Supplementary Fig. 7 an overlay between the present E. coli CSU model and the region of a previous model of the T. maritima CSU. The latter was originally modelled by Keith Cassidy who also did the modelling here, and was based on cryo-ET data from E. coli (PDB 3JA6), the only other published coordinates of a CSU complex. Considering that the two models have different sequences, T. maritima receptors are generally longer and the T. maritima model does not contain a membrane-spanning region, we have avoided using a single-valued metric to assess the goodness-of-fit (e.g., the root-mean-squared displacement between models). In general, such a number does not capture usefully the similarities and differences between large, multi-protein complexes. Rather, we believe that the presently provided overlay offers a clear, qualitative comparison appropriate for the resolution of the cryo-ET data the level of the discussion regarding the model in the present manuscript. In particular, Supplementary Fig. 7 illustrates the significant overlap between CheA, CheW, and the receptor tips in the CSU baseplate, while Fig. 4 highlights the substantially extension of the CSU model made in the present study.

3. The map does not look like a 16 Å resolution map to me. FSC curve would be helpful. Reading the methods section, it seems as if all subtomograms were aligned together but averaged in two halves for the FSC measurement. In this case, please use the 0.5 criterion for resolution estimation. You can use 0.143 if the maps have been aligned and averaged completely independently (see Scheres and Chen, Nat Methods).

As specified in the Methods section, the FSC of the half-maps inside a mask, containing three CSUs and no membrane, drops below 0.5 and 0.143 at spatial frequencies corresponding to resolution estimates of 18 Å and 16 Å respectively. We now show the FSC plot at Supplementary Fig. 4. The maps were subsequently subjected to localised resolution estimation in RELION with a sampling rate of 20 Å and locally filtered to the estimated resolution of the map, which is probably the reason why the reviewer has an impression that the map does not look like a 16 Å resolution map – the resolution is indeed quite variable at different areas of the map (between ~15-30 Å), as we now show in Supplementary Fig. 4.

To satisfy the concerns of the reviewer, we now go even further in the local resolution analysis and present a way to evaluate local resolution distribution in each major region of the CSU cryo-ET map based on our molecular model. The method is described in the text (lines 208-216) and in the Fig. 4c. This representation greatly facilitates the interpretation of the map. Furthermore the same strategy can be used for analysis of structures of all multisubunit complexes, provided the availability of model pDBs fitted inside the cryo-EM density of interest.

4. Concerning the new minicell line, I cannot see the loading controls in the gel although the authors claim to use one in the methods section. Could you show these please?

The markers can be seen in the Source.xlsx file of the Fig. 3 where, according to the Nature policy, we show the entire gel.

5. Additionally one main point of the manuscript is that their new cell line is an useful tool to study chemosensory arrays because those minicells are very small and have to 50% those arrays. What is "n" here? How many cells were imaged? I am unconvinced that this is a huge improvement. The authors should try and focus on the biology revealed by the structure.

It is true that the CSU reconstruction and derived model in the present study is based on 6 tomograms only. However we also collected other datasets under various different conditions, totalling >200 tomograms. In particular, we are in the process of analysing a much larger dataset of more than 100 tilt series, and observe the presence of the array in about half of the reconstructed tomograms.

Indeed, because previously published data on other types of minicells is not publically available, stricto sensu we cannot distinguish whether the superior quality of our cryo-ET maps of the CSU is due to some specific properties of our minicell strain (as analysed in the first half of the present manuscript) or to our image analysis strategy.

6. Additionally there are quite a few typos or inconsistencies in the format between the sections. For example “chemoecetpor” in Line591, switching between ml and mL or “x g” and “g”, a total dose as e- per Å² per sec instead of e- per Å². Could the authors please standardise the manuscript.

We have now corrected the typos throughout the manuscript.

Reviewer #4 (Remarks to the Author):

This is a potentially landmark study. However, I have a lot of concerns with the brevity of the all-atoms model. I do not think it is ‘wrong’, it appears to be a nice work actually, from a technical point of view, but I do not see its point either, because there is no discussion of the model at all or comparison of the model with previous published models of the (partial, admittedly) chemotaxis apparatus models.

I understand that the all-atoms model is ‘to be published later’ in details, but that statement is not enough in my opinion to justify the publication of the model. Could the authors at least comment on whether or not their model is compatible with previous published dynamics mechanism of chemotaxis, some of them, apparently, with co-authorship by some of the author(s) of the present manuscript? For instance:

<https://www.ncbi.nlm.nih.gov/pmc/articles/PMC3123975/> (Tar TM)

<https://elifesciences.org/articles/08419> (3JA6)

<https://journals.plos.org/ploscompbiol/article?id=10.1371/journal.pcbi.1002685> (Hall Tar)

http://chemotaxis.biology.utah.edu/Parkinson_Lab/publications/PDFs/Ortega%20et%20al.,%202013.pdf (Davi PHE)

We thank the reviewer for the positive assessment of the quality of the modelling work. We have taken several steps to improve our discussion surrounding the model to better highlight its utility in the present study and further justify its publication. First and foremost, we wish to emphasize that the CSU model provides several structural insights that could not be inferred from interpretations of the map alone. We have altered the discussion in several places to better highlight these, including enabling (1) an assessment of the local variability in map resolution (lines 208-216), (2) the first characterization of the periplasmic organization of the array (lines 219-230), (3) construction of a clash-free, all-atom model satisfying the structural constraints of the cryo-ET data (lines 230-233), and (4) isolation of novel density between the CheA.P4 domains, likely corresponding to CheA.P1/P2 (lines 246-254). Moreover, the present study represents the first use of molecular modelling to combine existing structural data from a wide range of experimental sources into a single, self-consistent model of the complete CSU complex. As such, we feel the CSU model is of significant value to the field as a proof of concept and sets the stage for much subsequent experimental/computational work that will provide novel mechanistic insights.

*As requested by the reviewer we have also provided a comparison between the present CSU model and a previously published model. This is included Fig. 4b and Supplementary Fig. 7, which show an overlay between the present CSU model and a previous model of the *T. maritima* CSU (PDB 3JA6, Cassidy et al., 2015), the only other published coordinates of a CSU complex. In particular, Supplementary Fig. 7 illustrates the significant overlap between CheA, CheW, and the receptor tips in the CSU baseplate, while Fig. 4b highlights the substantially extension of the CSU model made in the present study. Moreover, we wish to note that the CSU model makes use of many of the same structural data utilized in the previous studies highlighted by the reviewer, including high resolution structures PDB 1QU7 (Ortega et al., 2013; Cassidy et al., 2015; Hall et al., 2012), PDB 2D4U (Hall et al. 2012), PDB 4JPB (Cassidy et al., 2015), as well as Tar disulfide cross-linking data (Park et al., 2011). Unfortunately, however, it is beyond the scope of the present study to assess how the dynamics of the CSU model might compare with the previously published ‘dynamics mechanisms of chemotaxis’. To do so would require long-timescale, all-atom simulations of the CSU model (>3.5 million atoms with solvent). While feasible technically, these simulations will be extremely intensive from a computational perspective and require months of runtime on large computer clusters. Thus, additional studies will be required to more fully investigate such topics.*

reviewers' Comments:

Reviewer #1:

Remarks to the Author:

The authors have addressed my concerns satisfactorily.

Reviewer #2:

Remarks to the Author:

The authors have addressed by concerns satisfactorily in the revision. Nice work!

Reviewer #3:

Remarks to the Author:

The authors have answered all my concerns. I recommend publication of the manuscript, which is much improved in my opinion.

One small request - could the authors please report the isosurface threshold levels in sigma values please (reported in Coot for example)? The absolute number is map dependent and not helpful.

Reviewer #4:

Remarks to the Author:

I thank the authors for including (some) more discussion on the model itself, making the modeling part less/not disconnected from the rest of the narrative. I am fine with the modifications.

I understand that the model is more of an illustration of the structure than the basis for a complete characterization of the structure-function-dynamics relationship. I regret that a bit, as I am genuinely impressed by the scale and the quality of the modeling. I I am very much looking forward to a manuscript that will focus on the system's structure-function-dynamics, and comparison with previous publications on chemotaxis systems' dynamics.

REVIEWERS' COMMENTS:

Reviewer #1 (Remarks to the Author):

The authors have addressed my concerns satisfactorily.

Reviewer #2 (Remarks to the Author):

The authors have addressed by concerns satisfactorily in the revision. Nice work!

Reviewer #3 (Remarks to the Author):

The authors have answered all my concerns. I recommend publication of the manuscript, which is much improved in my opinion.

One small request - could the authors please report the isosurface threshold levels in sigma values please (reported in Coot for example)? The absolute number is map dependent and not helpful.

In EM, the sigma values depend on the size of the box and are therefore map-dependent per se. We feel that we already satisfied the Reviewer's concern by providing the two separate thresholds used for the periplasmic and intracellular domains. Indeed, the most important is that (i) the values for the periplasmic and the intracellular domains are different because the intracellular domains are better defined, as extensively discussed in the manuscript, and (ii) an interested reader can download the map and use the values that we currently provide in order to reproduce our visualisation in Chimera.

Reviewer #4 (Remarks to the Author):

I thank the authors for including (some) more discussion on the model itself, making the modeling part less/not disconnected from the rest of the narrative. I am fine with the modifications.

I understand that the model is more of an illustration of the structure than the basis for a complete characterization of the structure-function-dynamics relationship. I regret that a bit, as I am genuinely impressed by the scale and the quality of the modeling. I I am very much looking forward to a manuscript that will focus on the system's structure-function-dynamics, and comparison with previous publications on chemotaxis systems' dynamics.

We thank the Reviewers for their help with the improvement of the manuscript.